# High-Resolution Bioassay Profiling with Complemented Sensitivity and Resolution for Pancreatic Lipase Inhibitor Screening

**DOI:** 10.3390/molecules27206923

**Published:** 2022-10-15

**Authors:** Jingyi Jian, Jiaming Yuan, Yu Fan, Jincai Wang, Tingting Zhang, Jeroen Kool, Zhengjin Jiang

**Affiliations:** 1Institute of Traditional Chinese Medicine & Natural Products, College of Pharmacy/Guangdong Province Key Laboratory of Pharmacodynamic Constituents of TCM and New Drugs Research/International Cooperative Laboratory of Traditional Chinese Medicine Modernization and Innovative Drug Development of Ministry of Education (MOE) of China, Jinan University, Guangzhou 510632, China; 2Division of BioAnalytical Chemistry, Amsterdam Institute of Molecules, Medicines and Systems, Vrije Universiteit Amsterdam, 1081 HV Amsterdam, The Netherlands

**Keywords:** high-resolution bioassay profiling, magnetic beads-based ligand fishing, high-performance liquid chromatography, pancreatic lipase, inhibitor screening

## Abstract

How to rapidly and accurately screen bioactive components from complex natural products remains a major challenge. In this study, a screening platform for pancreatic lipase (PL) inhibitors was established by combining magnetic beads-based ligand fishing and high-resolution bioassay profiling. This platform was well validated using a mixture of standard compounds, i.e., (-)- epigallocatechin gallate (EGCG), luteolin and schisandrin. The dose–effect relationship of high-resolution bioassay profiling was demonstrated by the standard mixture with different concentrations for each compound. The screening of PL inhibitors from green tea extract at the concentrations of 0.2, 0.5 and 1.0 mg/mL by independent high-resolution bioassay profiling was performed. After sample pre-treatment by ligand fishing, green tea extract at the concentration of 0.2 mg/mL was specifically enriched and simplified, and consequently screened through the high-resolution bioassay profiling. As a result, three PL inhibitors, i.e., EGCG, (-)-Gallocatechin gallate (GCG) and (-)-Epicatechin gallate (ECG), were rapidly identified from the complex matrix. The established platform proved to be capable of enriching affinity binders and eliminating nonbinders in sample pre-treatment by ligand fishing, which overcame the technical challenges of high-resolution bioassay profiling in the aspects of sensitivity and resolution. Meanwhile, the high-resolution bioassay profiling possesses the ability of direct bioactive assessment, parallel structural analysis and identification after separation. The established platform allowed more accurate and rapid screening of PL inhibitors, which greatly facilitated natural product-based drug screening.

## 1. Introduction

Obesity is an increasing world health crisis with many potential and serious complications, e.g., hypertension, hyperlipemia and diabetes [1,2]. One of the most important drug targets of obesity is pancreatic lipase (PL). PL inhibitors can help to control obesity. Orlistat, as well as some other well-known obesity medicines, can inhibit PL [3,4]. However, many of these drugs may cause side effects, e.g., gastrointestinal adverse reactions. In this context, the discovery and development of novel PL inhibitors is of great significance.

Natural products have been constantly reported to exhibit PL inhibitory properties [5,6,7]. Many techniques and approaches have been developed and applied to screening bioactive compounds from complex natural products, which are mostly time-consuming, laborious and costly [8,9]. In this regard, bioassays integrating analytical chemistry have recently made considerable progress in rapid profiling and identification of bioactive compounds from natural extracts. For example, many technologies are dedicated to developing fast, sensitive and high-throughput screening methods for identification of bioactive compounds in complex mixtures, including pre-column ligand fishing approaches using ultrafiltration [10,11,12] or magnetic beads [13,14], on-column bioaffinity separations like cellular membrane chromatography [15,16], cellular membrane affinity chromatography [17,18], and post-column methods such as high-resolution bioassay profiling [19,20,21].

Among these methods, high-resolution bioassay profiling is advantageous in having direct parallel bioactivity and mass spectrometer (MS)-based identification potential of bioactive substances in complex mixtures, and it allows the implementation of straightforward plate reader-based bioassays. However, in terms of screening from natural extracts, this platform often suffers from the inefficiency in sensitivity for low-abundance components due to column capacity and separation ability of high-performance liquid chromatography (HPLC), as well as the inefficiency in resolution due to overlapping peaks in chromatograms. In our previous work, increased concentrations and multiple sample collections were used to overcome the inefficiency of sensitivity [20]. To enhance the resolution, both reversed-phase and hydrophilic interaction liquid chromatography were applied for analyzing natural products [20,22]. Nevertheless, these adjustments could increase the consumption of time, proteins and other solvents.

Another screening method is magnetic beads-based ligand fishing based on the affinity between immobilized target proteins and compounds. In our previous work, magnetic beads-based ligand fishing was successfully established for inhibitor screening of neuraminidase [23,24] and monoamine oxidase-A [25] from compound libraries and natural products. It is simple to prepare and fast to perform; however, the screening mechanism by affinity may sometimes cause false positive/negative results because of non-specific binding and disability in assessing the activity of samples.

In order to develop a fast-screening platform for pancreatic lipase inhibitors with higher sensitivity and resolution, we combined magnetic beads-based ligand fishing and high-resolution bioassay profiling for screening PL inhibitors. In this endeavor, false positive/negative results of ligand fishing can be eliminated by taking the advantage of high-resolution bioassay (direct parallel bioactivity). Meanwhile, using magnetic beads-based ligand fishing as sample pre-treatment, the complex mixtures from natural products can be simplified and compounds at low concentrations can be enriched. Eventually, this platform was successfully applied to screen PL inhibitors from green tea extract [5].

## 2. Results and Discussions

As shown in Figure 1, the proposed screening platform consists of two modules, i.e., magnetic beads-based ligand fishing and high-resolution bioassay profiling. At the start of the strategy, complex samples such as natural product extracts are pre-treated by the module of magnetic beads-based ligand fishing to enrich affinity binders and eliminate non-binders to increase the concentration of potential active compounds and avoid overlapping of non-binders in chromatogram. Then, the pre-treated samples are introduced to the module of high-resolution bioassay profiling to directly separate (by HPLC), identify (by MS) and assess (by bioassay and plate reader) the captured binders. This integrated platform is expected to enhance the sensitivity and resolution of the high-resolution bioassay profiling. The mixture of standard compounds was used to validate this platform. Green tea extract was finally screened and analyzed to discover potential PL inhibitors.

### 2.1. Validation of Ligand Fishing Using Pancreatic Lipase-Modified Magnetic Beads

Standard mixture A, consisting of the inhibitor luteolin and the negative control schisandrin, was used to perform ligand fishing via PL-modified magnetic beads. As shown in Figure 2, the chromatographic peaks corresponding to luteolin (retention time = 9.5 min) in E_A_1, 2, 3 represented the specific binders, where E_A_1 possessed the highest peak. In contrast, almost all the schisandrin remained in S_A_1 and proved no binding of schisandrin to PL. This result demonstrated that using ligand fishing, the compounds in the mixtures (S_A_0) without affinity to PL could be eliminated, as such a simplified sample (E_A_1) with affinity compounds could be prepared.

### 2.2. Validation of High-Resolution Bioassay Profiling

Next, the high-resolution bioassay profiling was established based on our previous work [20], which was adapted to PL in this work. The profiling was demonstrated using standard mixture B consisting of two inhibitors, i.e., (-)- epigallocatechin gallate (EGCG) (IC_50_ = 0.97 μM, Appendix A) and luteolin (IC_50_ = 30.57 μM, Appendix A), as well as the negative control schisandrin. As presented in Figure 3a, each point in the bioassay chromatograms represented the slope of the kinetic signals measured in a well in a 384-well plate plotted against the fractionation time of that well. The bioactivity chromatograms were obtained by plotting these bioassay points of the standard mixture. In all the bioactivity chromatograms, two negative peaks were observed resulting from the eluted inhibitors, i.e., EGCG and luteolin, which inhibited the activity of PL. This could thereby reduce the formation of the enzymatic fluorescent products, which in turn yielded reduced slopes and thus a lower signal on the y-axis. In addition, EGCG caused a stronger negative peak at the same concentrations than luteolin due to its higher activity. As expected, the non-bioactive compound schisandrin did not lead to a corresponding negative peak in the bioactive profiles. A clear concentration response effect was observed in the bioactivity chromatograms: the height of the two negative peaks increased with increasing concentrations of EGCG and luteolin.

### 2.3. Integration of Ligand Fishing and High-Resolution Bioassay Profiling

As shown in Figure 4a, by increasing the amount of PL immobilized on magnetic beads (PL-MBs) and sample solution, and by decreasing the volume of elution solvent (referred to as 3.6), the amount of EGCG and luteolin increased in E_B_1 (specific binders collected after elution) compared to those in S_B_0 (standard mixture B), which indicated an enrichment of the inhibitors by ligand fishing. As a result, the negative peaks in bioactivity chromatograms became larger and stronger. Figure 4b indicated an elimination of schisandrin in S_B_0 compared to E_B_1, showing that ligand fishing could simplify the mixtures by eliminating the compounds without affinity.

### 2.4. Screening PL Inhibitors from Green Tea Extract

The high-resolution bioassay profiling of green tea extracts without ligand fishing were illustrated in Figure 5. As the concentrations of green tea decreased (Figure 5b), three negative peaks become smaller (Figure 5a). Negative peak 2 in Figure 5a became unclear at the lowest concentration of 0.2 mg/mL, which revealed the sensitivity limit of the high-resolution bioassay profiling.

In addition, the mass spectra (Figure 6) showed that the chromatographic elutes corresponding to the negative peaks 1 and 3 contained pure active compounds with masses of 457.08 and 441.08, respectively. Based on the retention times (Figure 5b), the MS/MS data (Figure 7) and the previously published work [26,27], the negative peaks 1 and 3 corresponded to EGCG and (-)-Epicatechin gallate (ECG), respectively. However, the chromatographic peak corresponding to the negative peak 2 has two masses, i.e., 457.08 and 479.08, which indicated that one active compound could not be distinguished by the retention time. In other words, the resolution of the high-resolution bioassay profiling needed to be further improved in this case.

In order to improve sensitivity and avoid overlapping in the high-resolution bioassay profiling, ligand fishing using PL-MBs was applied to enrich and simplify complex samples. In Figure 8, the enriched and simplified sample of green tea extract (E1) was directed to the high-resolution bioassay profiling and compared with the original sample of green tea extract (S0). Due to the sample enrichment by ligand fishing, chromatographic elutes corresponding to the negative peaks 1, 2 and 3 in E1 increased (Figure 8b,c) and the negative peak 2 became more obvious (Figure 8a), suggesting higher sensitivity.

Moreover, comparing the mass spectra between S0 and E1 (Figure 9), the mass intensity increased in E1 for all the contents corresponding to the negative peaks 1, 2 and 3 except for the mass of 479.08. This demonstrated the enrichment of active compounds and elimination of inactive compounds (simplification) by ligand fishing. Based on the MS/MS data (Figure 7), the active compound 2 corresponding to the negative peak 2 was (-)-Gallocatechin gallate (GCG) with the mass of 457.08 [26,27].

In addition, some other compounds in E1 were also enriched by ligand fishing, as revealed by the enlargement of these chromatography peaks (Figure 8b). According to the individual high-resolution bioassay profiling for the green tea extracts at increasing concentrations (0.2, 0.5 and 1.0 mg/mL) (Figure 5a) and high-resolution bioassay profiling with enriched (E1) green tea extracts (Figure 8a), these compounds could be identified as non-specific binders as no negative peaks were observed in the bioassay profiles.

## 3. Materials and Methods

### 3.1. Chemicals and Reagents

Lipase from porcine pancreas, 4-methylumbelliferyl oleate, hydroxymethyl aminomethane hydrochloride (Tris-HCl), 4-morpholineethanesulfonic acid (MES), N-Hydroxysuccinimide (NHS) and 1-(3-Dimethylaminopropyl)-3-ethylcarbodiimide hydrochloride (EDC) were purchased from Sigma-Aldrich (Guangzhou, China). Formic acid (FA), (-)- epigallocatechin gallate (EGCG), luteolin, schisandrin and dimethyl sulfoxide (DMSO) were purchased from Aladdin Chemistry Co. (Shanghai, China). Merck (Darmstadt, Germany) provided HPLC-grade methanol (MeOH) and acetonitrile (ACN) deionized water was purified by a Milli-Q purification system (Guangzhou, China).

### 3.2. Sample Preparation of Green Tea Extracts

Dried green tea leaves (10 g) were smashed and sonicated (40 kHz, 250 W) in 100 mL water for 50 min at 40 °C using a KQ2200DV sonicator (Kunshan Ultrasonic Instruments, Kunshan, China). After filtration, the filtrate was extracted by petroleum ether and ethyl acetate in sequence. The ethyl acetate extract was dried, collected and stored at −20 °C prior to use.

### 3.3. PL Bioassay

PL was stored as lyophilized powder at 4 °C. 4-Methylumbelliferyl oleate (4-MUO) was dissolved in DMSO as 10 mM stock solution and stored at −20 °C. To initiate the bioassay, lipase was dissolved at room temperature to a final concentration of 1 μg/mL in Tris-HCL buffer (13 mM, pH 8.0). 4-MUO stock solution was diluted to 30 μM in Tris-HCL buffer. Regarding these solutions, 25 μL lipase solution (1 μg/mL) and 25 μL 4-MUO solution (30 μM) were added in sequence to the nanofractionated black 384-well plates by a Thermo Fisher Multidrop™ 384 Reagent Dispenser (Ermelo, The Netherlands). Next, each well plate was placed in a Biotek plate reader (Guangzhou, China) and measured in fluorescence mode at 360 nm excitation and 460 nm emission wavelengths in kinetic fashion for 60 min. Finally, the slopes of the resulting kinetic curves per well were calculated, which represented the enzymatic activity of lipase in each well. The resulting slopes were then normalized and plotted as bioactivity chromatograms by plotting the slopes versus time of each collected fraction using OriginPro 9.0 64-bit software (Version 9.0, OriginLab Corporation, Northampton, MA, USA). The slopes were normalized by dividing each slope value with the median of all the values obtained from a single plate-reader measurement.

### 3.4. Ligand Fishing Using PL Magnetic Beads

The PL-MBs were prepared as follows: 10 mg of BeaverBeads™ Mag COOH_2_ carboxyl-terminated magnetic beads (Beaverbio Co., Ltd., Suzhou, China) were first washed three times with 1 mL of MES buffer (100 mM MES pH = 5.0). The beads were then re-suspended in 1 mL MES buffer with EDC (5 mg/mL) and NHS (5 mg/mL) and rotated at 25 °C for 0.5 h. After magnetic separation, the beads were resuspended in 1 mL of 10 mg/mL PL solution (PB, 100 mM pH = 7.4) and rotated at 25 °C for 6 h. Finally, the beads were washed three times with 1 mL of PB buffer and stored at 4 °C.

In order to demonstrate this method, mixture A (luteolin + schisandrin) was set up and dissolved in PB at the concentration of 25 μM for each compound to perform ligand fishing. 2 mg PL-MBs were suspended in 200 μL mixture A solution (S_A_0) and incubated for 30 min at room temperature. After magnetic separation, the supernatant (S_A_1) was collected, and PL-MBs were washed by 200 μL PB three times to collect the supernatants of washing (W_A_1, 2, 3), followed by elution with 200 μL PB (25%, *v*/*v*) three times to collect the supernatants of elution (E_A_1, 2, 3). After collection, S0A and each supernatant were injected into an Agilent 1100 HPLC system (Agilent Technologies, Inc., Waldbronn, Germany) for analysis.

### 3.5. High-Resolution Bioassay Profiling for Screening PL Inhibitors

Liquid chromatography separation was carried out using an Agilent 1100 HPLC system. The sample injection volume was 10 μL. The separation was performed on a Waters sunfire C18 column (4.6 mm × 250 mm, 5 μm), with a flow rate of 0.5 mL/min. The mobile phase A was H_2_O with 0.1% FA and the mobile phase B was ACN with 0.1% FA. After chromatography separation, the eluent was spilt in a 2:1 ratio. The larger eluent portion was connected to a CTC-PAL autosampler (CTC Analytics AG, Zwingen, Switzerland) to perform nanofractionation with the fraction collection resolution of 8 sec onto Greiner black 384-well plates (Shanghai, China). The nanofractionated plates were then dried in a vacuum oven (Shanghai Xiaohan Industrial Development, Shanghai, China) prior to the subsequent bioassays. The smaller eluent portion was directed to DAD (254 nm) and AB SCIEX X500R Quadrupole-Time-of-Flight Mass Spectrometer (Q-TOF-MS/MS) equipped with an electrospray ionization (ESI) source (Redwood City, CA, USA) using SCIEX OS software ver. 1.5. MS operating conditions were as follows: curtain gas: 25 psi; ion source gas 1: 60 psi; ion source gas 2: 60 psi; ion source temperature: 600 °C; ion spray voltage: −4.5 or 5.5 kV. TOF-MS scan range: 100–1000 Da with a 0.25 s ion accumulation time and a collision energy of −10 or 10 V; 50–1000 Da (TOF-MS/MS scan range) with a 0.1 s accumulation time and a collision energy of −35 or 35 V. Mixture B (EGCG + luteolin + schisandrin), dissolved in PB at the concentrations of 25, 50 and 100 μM for each compound, was applied to validate the high-resolution bioassay profiling.

### 3.6. Demonstration of the Integrated Ligand Fishing and High-Resolution Bioassay Profiling

Mixture B (EGCG + luteolin + schisandrin) was dissolved in PB at the concentration of 25 μM. To achieve the enrichment of active compounds, a higher amount of PL-MBs (5 mg) and a lower volume of elution solvent (200 μL) were used for ligand fishing. First, 5 mg PL-MBs were suspended in 500 μL mixture B solution (S_B_0) and incubated for 30 min at room temperature. After magnetic separation, the supernatant (S_B_1) was collected, and PL-MBs were washed by 500 μL PB three times to collect the supernatants of washing (W_B_1, 2, 3) and eluted by 200 μL PB (25%, *v*/*v*) three times to collect the supernatants of elution (E_B_1, 2, 3). Next, S_B_0 and E_B_1 were injected into the high-resolution bioassay profiling for fractionation, bioassay and analysis.

### 3.7. Screening of PL Inhibitors from Green Tea Extract

Green tea extracts were dissolved in PB at the concentrations of 0.2, 0.5 and 1.0 mg/mL, and screened directly by the high-resolution bioassay profiling. For comparison, 0.2 mg/mL of green tea solution was screened by the integrated ligand fishing and high-resolution bioassay profiling using the same procedure as described above. S0 and E1 were injected into high-resolution bioassay profiling for fractionation, bioassay and analysis.

## 4. Conclusions

In this study, a high-resolution bioassay profiling was established for screening PL inhibitors. In order to enhance the sensitivity and resolution of the bioassay profiling in case of complex samples such as natural products, ligand fishing using magnetic beads immobilized with PL was applied as pre-treatment to enrich and simplify complex samples using more PL-MBs and a smaller volume of elution solvent. Green tea extract was screened for PL inhibitors employing this integrated platform. Three active compounds were identified as (-)-Epigallocatechin gallate (EGCG), (-)-Gallocatechin gallate (GCG) and (-)-Epicatechin gallate (ECG) using a relatively lower concentration of green tea extract.

Notably, ligand fishing successfully enhanced the sensitivity and resolution of the high-resolution bioassay profiling by enriching low-abundance active compounds and eliminating the compounds without affinity. Additionally, without using blank control, the non-specific binders could also be directly eliminated by the high-resolution bioassay profiling because of an absence of corresponding negative peaks in the bioassay profiles. The integration of ligand fishing with high-resolution bioassay profiling could overcome some inherent deficiencies, such as non-specific binding, insufficient sensitivity for weak and low-abundance inhibitors, overlapping of peaks, etc. This provided a fast-screening method for active compounds from complex matrices with complemented sensitivity and high-resolution.

## Figures and Tables

**Figure 1 molecules-27-06923-f001:**
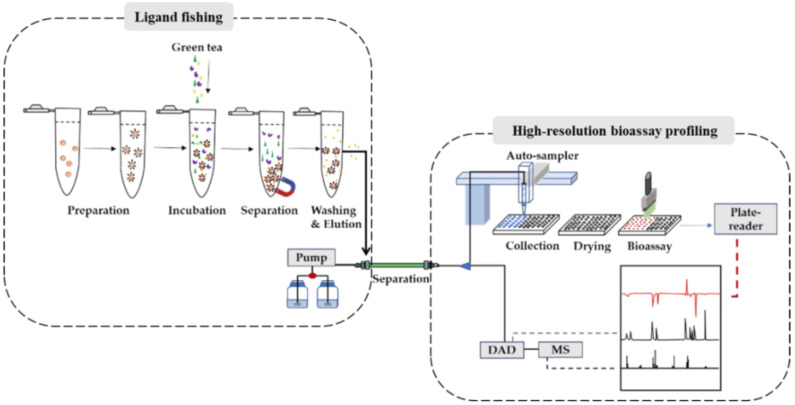
Concept of the integrated ligand fishing and high-resolution bioassay profiling.

**Figure 2 molecules-27-06923-f002:**
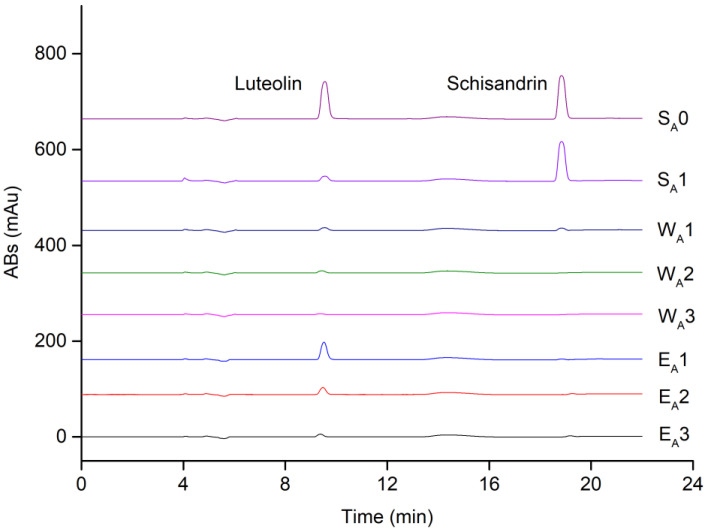
Chromatograms of ligand fishing for mixture A at 254 nm. S_A_0 corresponded to standard mixture A and S_A_1 corresponded to its supernatant after incubation. W_A_1, 2, 3 indicated the non-specific binders collected after washing and E_A_1, 2, 3 indicated the specific binders collected after elution, with “1, 2, 3” indicating the times for washing or elution.

**Figure 3 molecules-27-06923-f003:**
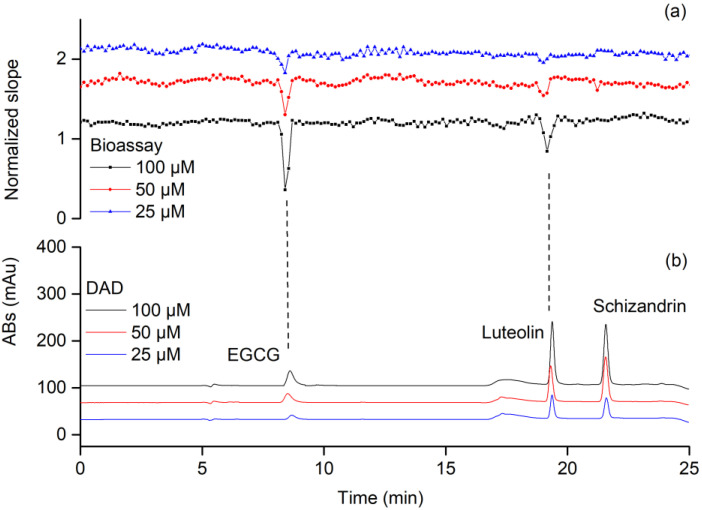
High-resolution bioassay profiling for mixture B. (**a**) Bioactivity chromatograms obtained from analyzing the mixture at the concentrations of 25, 50 and 100 μM. (**b**) Parallelly obtained diode-array detector (DAD) chromatogram at 254 nm.

**Figure 4 molecules-27-06923-f004:**
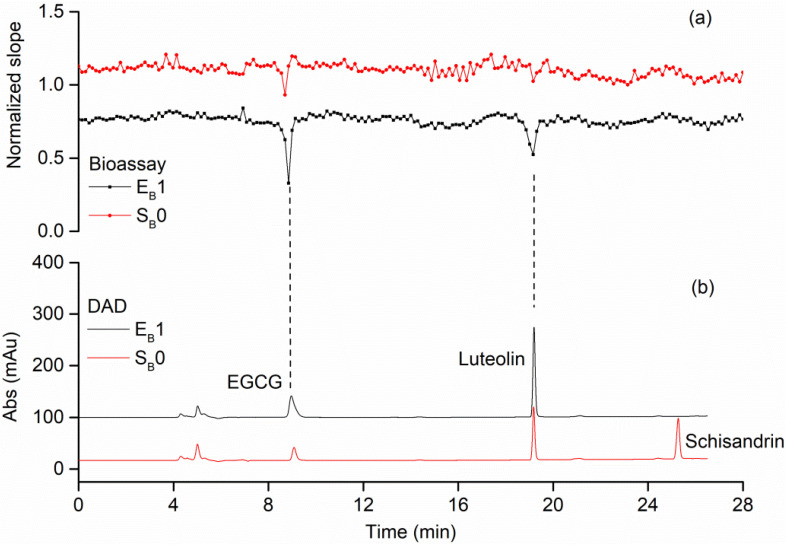
High-resolution bioassay profiling for mixture B after sample simplification and enrichment by magnetic ligand fishing. (**a**) Bioactivity chromatograms obtained from analyzing the solutions of S_B_0 (standard mixture B) and E_B_1 (specific binders of mixture B). (**b**) Parallelly obtained DAD chromatogram at 254 nm of the solutions of S_B_0 and E_B_1.

**Figure 5 molecules-27-06923-f005:**
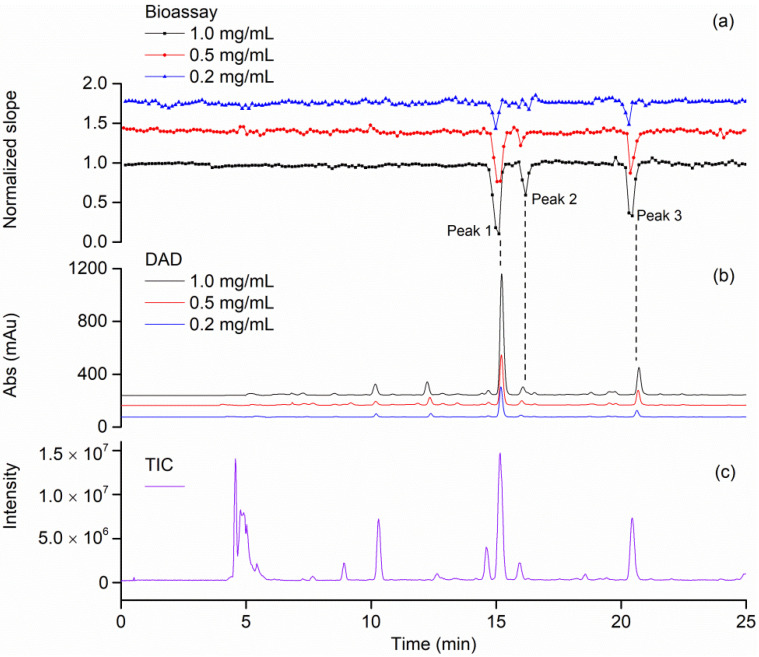
High-resolution bioassay profiling for green tea extract. (**a**) Bioactivity chromatograms obtained from analyzing the green tea extract solutions at the concentrations of 0.2, 0.5 and 1.0 mg/mL. (**b**) Parallelly obtained DAD chromatogram at 254 nm of the solutions at the concentrations of 0.2, 0.5 and 1.0 mg/mL. (**c**) Total ion chromatograms (TIC) of green tea extract.

**Figure 6 molecules-27-06923-f006:**
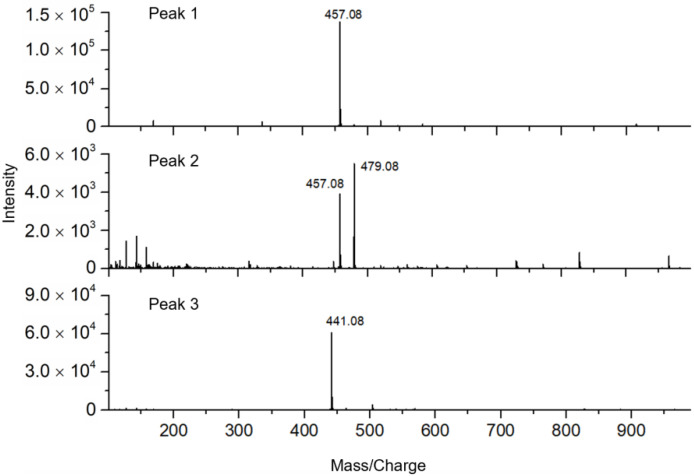
Mass spectra of potential bioactive contents in green tea extract, corresponding to the negative peaks 1, 2 and 3 in Figure 5.

**Figure 7 molecules-27-06923-f007:**
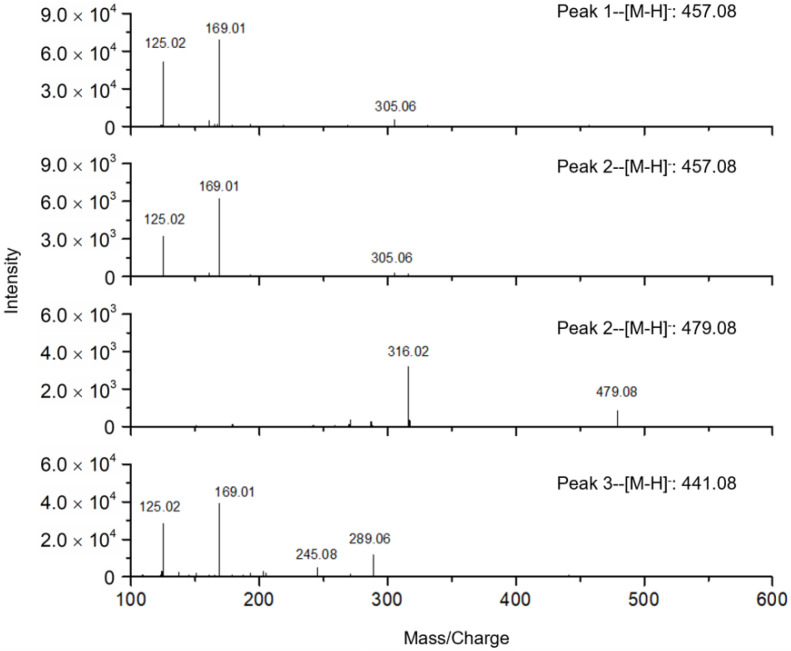
MS/MS spectra of potential bioactive contents in green tea extract, corresponding to the negative peaks 1, 2 and 3 in Figure 5.

**Figure 8 molecules-27-06923-f008:**
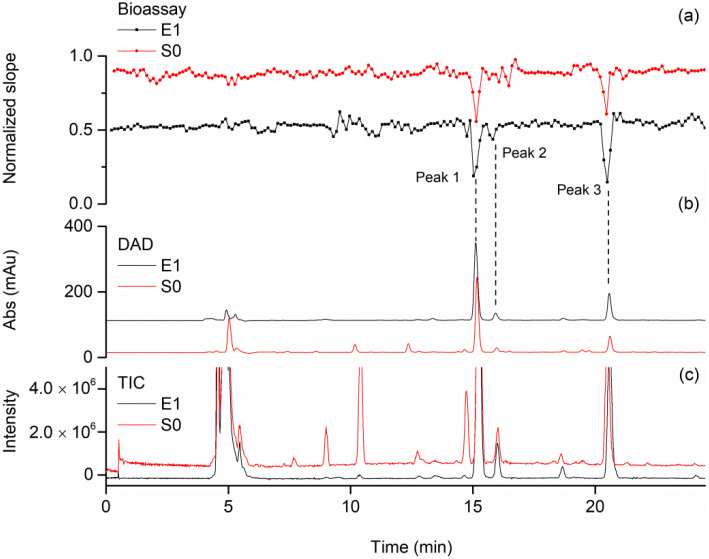
High-resolution bioassay profiling for green tea extract after sample simplification and enrichment by magnetic ligand fishing. (**a**) Bioactivity chromatograms obtained from analyzing the solutions of S0 (green tea extracts) and E1 (specific binders of green tea extract). (**b**) Parallelly obtained DAD chromatogram at 254 nm of the solutions of S0 and E1. (**c**) Total ion chromatograms (TIC) of green tea extract of S0 and E1.

**Figure 9 molecules-27-06923-f009:**
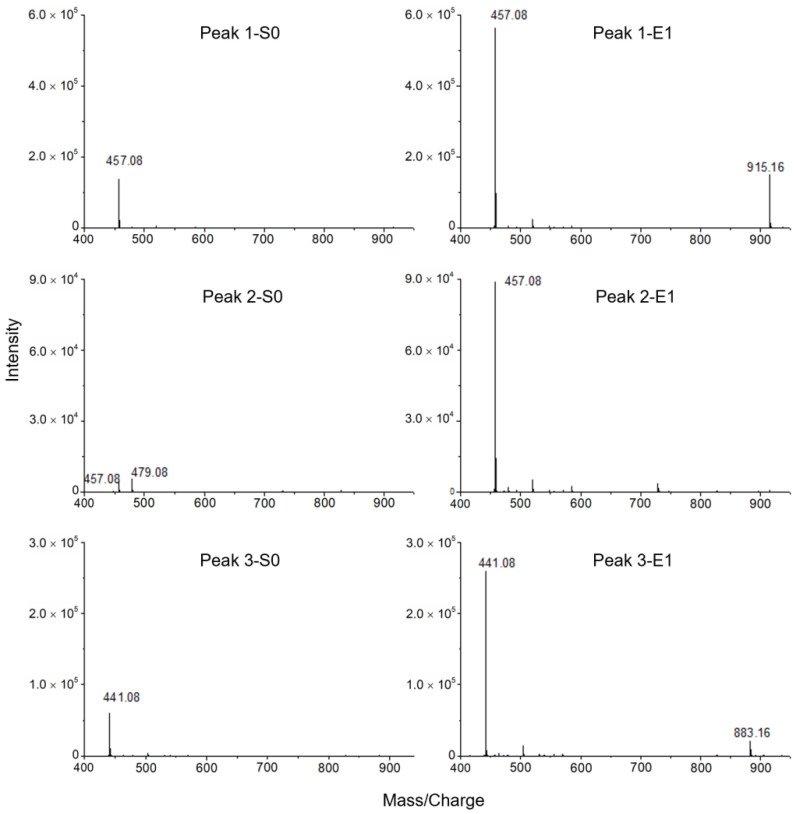
Mass spectra of potential bioactive contents from green tea extract in S0 and E1.

## Data Availability

Not applicable.

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
