# Peer review of "High-Resolution Bioassay Profiling with Complemented Sensitivity and Resolution for Pancreatic Lipase Inhibitor Screening"

_molecules, 2022, doi:10.3390/molecules27206923_

Round 1

Reviewer 1 Report

The study entitled “High-Resolution Bioassay Profiling with Complemented Sensitivity and Resolution for Pancreatic Lipase Inhibitor Screening” is a very interesting, topical one, a manuscript that I have gone through with pleasure.

It is an easy manuscript to read, very clear and explicit. However, I would have liked it to have shown more clearly the aim of the study and the conclusions the conclusions section should show what exactly was identified in the manuscript, I do not consider it necessary to describe the procedure again.

Good luck! 

Author Response

Dear editor and reviewer:

       Thank you very much for giving us an opportunity to revise our manuscript. We appreciate referees very much for their positive and constructive comments and suggestions on our manuscript. We have studied the referees’ comments carefully and made revisions accordingly. We hope the revision will improve the quality of the manuscript. The following is a list of the detailed revisions:

Answers to Reviewer 1 

      Thanks for your comments. The aim of this study has been supplemented in the last paragraph of introduction. Besides, in the conclusion, EGCG, GCG and ECG were identified (line 304&305), and their chemical names were added in the revised conclusion. The conclusions section has been re-written according to your suggestion. The changes in the revised manuscript have been highlighted.

Best regards,

Zhengjin Jiang

Reviewer 2 Report

Comments to Authors

This study aimed to describe a fast-screening platform for pancreatic lipase (PL) inhibitors by combining magnetic beads-based ligand fishing and high-resolution bioassay profiling. The work discusses an interesting method for pancreatic lipase inhibitors screening. However, the manuscript needs to be improved, some of the main points are as follows:

Page 1, line 21: the concentrations of 1.0, 0.5 and 0.2 mg/mL .....correct to the concentrations of 0.2, 0.5 and 1.0 mg/mL.

Page 2, line 63: to complement.... correct to to overcome

Page 2, line 64: to complement.... correct to to enhance

Page 2, line 66: How these adjustments could increase the proteins? Please explain

Page 2, line 75: In order to complement .....correct to in order to enhance

Page 2, line 78: Please re-write the sentence as ... can be eliminated by taking the advantage of high resolution bioassay (direct parallel bioactivity).

Page 2, line 92: is expected to complement the sensitivity ... correct to is expected to enhance the sensitivity...

Page 4, line 127: the sentence should be modified as: A clear concentration response effect was observed in the bioactivity chromatograms.

Page 4, line 132: simplify... do you mean modify the mixture?

Page 5, line 150: exhibited .....correct to illustrated

Page 5, line 163: (ECG) respectively ....correct to (ECG), respectively

Page 7, line 175: simplify... do you mean modify?

Page 10, line 217: the sentence should be correct to .. The ethyl acetate extract was dried, collected and stored at – 20 °C prior to use.

Page 10, line 220: Please mention the complete term for 4-MUO

Page 10, line 220-223: why did the authors prepare two concentrations from 4-MUO solution (10 mM and 30 mM. Additionally, the method should be re-written to be more clear.

Page 10, line 227: was measured ...correct to measured (remove was)

Page 11, line 280: the sub-title should be corrected to ... Screening of PL inhibitors from green tea extract.

Page 11, line 289: In order to complement correct to... In order to enhance

·        A list of abbreviations should be added.

Author Response

Dear editor and reviewer:

      Thank you very much for giving us an opportunity to revise our manuscript. We appreciate referees very much for their positive and constructive comments and suggestions on our manuscript. We have studied the referees’ comments carefully and made revisions accordingly. We hope the revision will improve the quality of the manuscript. The following is a list of the detailed revisions:

Answers to Reviewer 2

(1) Page 1, line 21: the concentrations of 1.0, 0.5 and 0.2 mg/mL .....correct to the concentrations of 0.2, 0.5 and 1.0 mg/mL.

Answer: Thanks for your advice. The manuscript has been revised according to your suggestion. The changes in the revised manuscript have been highlighted.

(2) Page 2, line 63: to complement.... correct to to overcome

Answer: Thanks for your advice. The manuscript has been revised according to your suggestion. The changes in the revised manuscript have been highlighted.

(3) Page 2, line 64: to complement.... correct to to enhance

Answer: Thanks for your advice. The manuscript has been revised according to your suggestion. The changes in the revised manuscript have been highlighted.

(4) Page 2, line 66: How these adjustments could increase the proteins? Please explain

Answer: In regular high-resolution bioassay profiling, only one mode of chromatography is applied for analyzing and subsequent bioassay. In order to overcome the efficiency of resolution, this experiment is performed separately using reversed-phase and hydrophilic interaction liquid chromatography, which means two bioassays are implemented so that the consumption of proteins in bioassay is doubled.

(5) Page 2, line 75: In order to complement ...correct to in order to enhance

Answer: Thanks for your advice. The manuscript has been revised according to your suggestion. The changes in the revised manuscript have been highlighted.

(6) Page 2, line 78: Please re-write the sentence as ... can be eliminated by taking the advantage of high resolution bioassay (direct parallel bioactivity).

Answer: Thanks for your advice. The sentence has been re-written according to your suggestion. The changes in the revised manuscript have been highlighted.

(7) Page 2, line 92: is expected to complement the sensitivity ... correct to is expected to enhance the sensitivity...

Answer: Thanks for your advice. The manuscript has been revised according to your suggestion. The changes in the revised manuscript have been highlighted.

(8) Page 4, line 127: the sentence should be modified as: A clear concentration response effect was observed in the bioactivity chromatograms.

Answer: Thanks for your advice. The sentence has been re-written according to your suggestion. The changes in the revised manuscript have been highlighted.

(9) Page 4, line 132: simplify... do you mean modify the mixture?

Answer: You probably referred to the line 141&175. After ligand fishing, compounds without affinity to pancreatic lipase in a mixture such as schisandrin in standard mixture A&B and non-specific binders in green tea extracts were washed out. The obtained supernatants of E1, EA1, EB1 with remaining affinitive compounds could be considered as simplified mixture.

(10) Page 5, line 150: exhibited .....correct to illustrated

Answer: Thanks for your advice. The manuscript has been revised according to your suggestion. The changes in the revised manuscript have been highlighted.

(11) Page 5, line 163: (ECG) respectively ....correct to (ECG), respectively

Answer: Thanks for your advice. The manuscript has been revised according to your suggestion. The changes in the revised manuscript have been highlighted.

(12) Page 7, line 175: simplify... do you mean modify?

Answer: Please refer to the answer in (9).

(13) Page 10, line 217: the sentence should be correct to .. The ethyl acetate extract was dried, collected and stored at – 20 °C prior to use.

Answer: Thanks for your advice. The sentence has been revised according to your suggestion. The changes in the revised manuscript have been highlighted.

(14) Page 10, line 220: Please mention the complete term for 4-MUO

Answer: Thanks for your advice. The complete term of 4-MUO has been added according to your suggestion. The changes in the revised manuscript have been highlighted.

(15) Page 10, line 220-223: why did the authors prepare two concentrations from 4-MUO solution (10 mM and 30 mM. Additionally, the method should be re-written to be more clear.

Answer: Thanks for your advice. It should be 10 mM and 30 μM. 10 mM of 4-Methylumbelliferyl oleate (4-MUO) was prepared as stock solution with high concentration. 30 μM of 4-MUO was used in bioassay. The method has been re-written according to your suggestion. The changes in the revised manuscript have been highlighted.

(16) Page 10, line 227: was measured ...correct to measured (remove was)

Answer: Thanks for your advice. The sentence has been revised according to your suggestion. The changes in the revised manuscript have been highlighted.

(17) Page 11, line 280: the sub-title should be corrected to ... Screening of PL inhibitors from green tea extract.

Answer: Thanks for your advice. The sentence has been revised according to your suggestion. The changes in the revised manuscript have been highlighted.

(18) Page 11, line 289: In order to complement correct to... In order to enhance

Answer: Thanks for your advice. The sentence has been revised according to your suggestion. The changes in the revised manuscript have been highlighted.

(19) A list of abbreviations should be added.

Answer: Thanks for your advice. List of abbreviations was added at the end of the manuscript according to your suggestion. The changes in the revised manuscript have been highlighted.

Best regards,

Zhengjin Jiang
